# Retrospective Case-Control Study of 2017 G2P[4] Rotavirus Epidemic in Rural and Remote Australia

**DOI:** 10.3390/pathogens9100790

**Published:** 2020-09-26

**Authors:** Bianca F. Middleton, Margie Danchin, Helen Quinn, Anna P. Ralph, Nevada Pingault, Mark Jones, Marie Estcourt, Tom Snelling

**Affiliations:** 1Global and Tropical Health, Menzies School of Health Research, Charles Darwin University, Darwin 0810, Australia; Anna.Ralph@menzies.edu.au (A.P.R.); tom.snelling@sydney.edu.au (T.S.); 2Division of Women, Children and Youth, Royal Darwin Hospital, Darwin 0810, Australia; 3Department of Paediatrics, University of Melbourne, Melbourne 3052, Australia; margie.danchin@rch.org.au; 4Murdoch Children’s Research Institute, Melbourne 3052, Australia; 5Department of General Medicine, Royal Children’s Hospital, Melbourne 3052, Australia; 6The National Centre for Immunisation Research and Surveillance (NCIRS), The Children’s Hospital at Westmead, Sydney 2145, Australia; helen.quinn@health.nsw.gov.au; 7Faculty of Medicine and Health, Westmead Clinical School, The University of Sydney, Westmead 2145, Australia; 8Division of Medicine, Royal Darwin Hospital, Darwin 0810, Australia; 9Department of Health Western Australia, Communicable Disease Control Directorate, Perth 6004, Australia; nevada.pingault@health.wa.gov.au; 10Health and Clinical Analytics, School of Public Health, The University of Sydney, Sydney 2006, Australia; mark.jones1@sydney.edu.au (M.J.); marie.estcourt@sydney.edu.au (M.E.); 11Wesfarmers Centre for Vaccine and Infectious Diseases, Telethon Kids Institute, Perth 6009, Australia; 12School of Public Health, Curtin University, Perth 6102, Australia

**Keywords:** rotavirus, rotavirus vaccines, vaccine effectiveness, case control

## Abstract

Background: A widespread G2P[4] rotavirus epidemic in rural and remote Australia provided an opportunity to evaluate the performance of Rotarix and RotaTeq rotavirus vaccines, ten years after their incorporation into Australia’s National Immunisation Program. Methods: We conducted a retrospective case-control analysis. Vaccine-eligible children with laboratory-confirmed rotavirus infection were identified from jurisdictional notifiable infectious disease databases and individually matched to controls from the national immunisation register, based on date of birth, Aboriginal status and location of residence. Results: 171 cases met the inclusion criteria; most were Aboriginal and/or Torres Strait Islander (80%) and the median age was 19 months. Of these cases, 65% and 25% were fully or partially vaccinated, compared to 71% and 21% of controls. Evidence that cases were less likely than controls to have received a rotavirus vaccine dose was weak, OR 0.79 (95% CI, 0.46–1.34). On pre-specified subgroup analysis, there was some evidence of protection among children <12 months (OR 0.48 [95% CI, 0.22–1.02]), and among fully vs. partially vaccinated children (OR 0.65 [95% CI, 0.42–1.01]). Conclusion: Despite the known effectiveness of rotavirus vaccination, a protective effect of either rotavirus vaccine during a G2P[4] outbreak in these settings among predominantly Aboriginal children was weak, highlighting the ongoing need for a more effective rotavirus vaccine and public health strategies to better protect Aboriginal children.

## 1. Introduction

Rotavirus is a leading cause of severe dehydrating diarrhoeal illness in children and continues to be responsible for the deaths of 118,000 to 183,000 children every year [1]. Many of these deaths occur in resource-poor settings [2].

In 2006, two oral rotavirus vaccines, Rotarix and RotaTeq, were licensed for use and in 2009 the World Health Organization endorsed their use globally [3]. Subsequent epidemiological studies have confirmed a strong protective effect of vaccination on rotavirus morbidity in high- and upper middle-income countries (vaccine efficacy [VE] >84%) [2]. However, in low-income countries, despite a large reduction in the absolute number of cases of gastroenteritis, measured vaccine efficacy has been lower (45–57%) and in some settings there is evidence of decreased protection in the second year of life [2,4,5,6].

The incorporation of rotavirus vaccines into the Northern Territory immunisation schedule in 2006 and then into the Australian National Immunisation Program (NIP) in 2007, resulted in a substantial and sustained decrease in rotavirus hospitalisations [7]. However, among Aboriginal and Torres Strait Islander children living in the hyperendemic settings of rural and remote Australia, the decrease in rotavirus hospitalisation was less dramatic and not sustained, with Aboriginal children living in the Northern Territory (NT) remaining more than 20 times more likely to be hospitalised with rotavirus than their non-Aboriginal counterparts [7]. An early vaccine effectiveness study in this setting also suggested reduced effectiveness against heterotypic strains and poor protection in the second year of life [8].

In 2017, an epidemic of G2P[4] rotavirus arose in the Northern Territory and subsequently spread to adjoining rural and remote regions of Western Australia (WA). These two jurisdictions cover a large geographic area which is sparsely populated; they have a higher proportion of resident Aboriginal and Torres Strait Islander people, many of whom live in rural and remote communities. The rotavirus epidemic occurred at a time when the Northern Territory exclusively administered Rotarix and Western Australia exclusively administered RotaTeq as part of the jurisdictional implementation of the NIP. We evaluated the protective effectiveness of both vaccines in these high-burden settings, ten years after the incorporation of rotavirus vaccines into the NIP.

## 2. Materials and Methods

### 2.1. Study Setting

The Alice Springs and Barkly regions of the Northern Territory, and the Kimberley, Pilbara and Goldfields regions of Western Australia are large but sparsely populated administrative health regions. Ranging from the semi-arid south, to the arid center and tropical north, these five regions encompass more than 2,500,000 km^2^, but are home to a combined total of just 174,000 people [9]. Children aged <5 years represent between 7–9% of the population, and between 5 and 41% of the population in each of these regions identify as being Aboriginal and/or Torres Strait Islander (hereafter respectfully referred to as ‘Aboriginal’) [9]. Many of these children live in towns or small remote communities. Rotarix and RotaTeq rotavirus vaccines have been licensed for use in Australia since June 2006. The Northern Territory immunisation program has funded the administration of Rotarix exclusively since October 2006. The Western Australian immunisation program funded the administration of Rotarix from July 2007 to June 2009, RotaTeq from July 2009 to June 2017, and Rotarix from July 2017.

### 2.2. Study Design

We conducted a retrospective, population-based, case control study of children age-eligible for at least 1 dose of rotavirus vaccine (those born after the introduction of Rotarix rotavirus vaccine to the NT schedule—after 1 July 2006 and aged ≥6 weeks, and those born after the introduction of RotaTeq rotavirus vaccine to the WA schedule—after 1 May 2009 and aged ≥6 weeks) who had laboratory positive and notified rotavirus infection during the 2017 G2P[4] rotavirus epidemic in the NT and WA. Cases were individually matched to controls sampled from the national immunisation register. As a secondary analysis, we also compared cases who were age-eligible for full rotavirus vaccination (those born after 1 July 2006 and aged ≥24 weeks in the NT and those born after 1 May and aged ≥32 weeks in WA) with un-matched control children diagnosed with non-rotavirus gastrointestinal infections sampled from disease notification registers.

### 2.3. Data Sources

Rotavirus is a notifiable disease in the NT and WA. Data regarding rotavirus cases and disease register controls were ascertained from the two jurisdictional-based notifiable infectious disease databases—The Northern Territory Notifiable Disease System (NTNDS) managed by the NT Centre for Disease Control, and the Western Australian Notifiable Infectious Disease Database (WANIDD) managed by the WA Department of Health.

To estimate baseline vaccine coverage in the case-referent population, matched population controls were sampled from the Australian Immunisation Register (AIR), a comprehensive population-based register which contains vaccination data for all children registered with Australia’s universal health insurance scheme, Medicare (~99% of the population).

### 2.4. Participants

#### 2.4.1. Population-Based Analysis

Rotavirus cases were vaccine-eligible children aged ≥6 weeks with laboratory positive and notified rotavirus infection between 1 March and 30 June 2017. Cases were drawn from the Alice Springs and Barkly regions of the NT, and the Kimberley, Pilbara and Goldfields regions of WA. To be vaccine-eligible, children had to be born on or after 1 July 2006 in the NT (for Rotarix) and on or after 1 May 2009 in WA (for RotaTeq).

De-identified population controls were selected from the Australian Immunisation Register and matched to each case by date of birth (±14 days), Aboriginal status and location of residence (listed residential postcode within either the Alice Springs, Barkly, Pilbara, Goldfields or Kimberley regions). Up to 10 eligible controls were randomly selected for each case.

#### 2.4.2. Disease Register Analysis

Rotavirus cases were selected as above, but because individual matching was not feasible, the analysis was restricted to children old enough to be fully vaccinated: age ≥24 weeks (for Rotarix) in the NT and ≥32 weeks (for RotaTeq) in WA.

Disease register controls were vaccine-eligible children (aged ≥24 weeks or ≥32 weeks in the NT and WA respectively), with microbiologically confirmed, non-rotavirus and non-vaccine preventable, notifiable gastrointestinal infections, notified between 1 January and 31 December 2017. Controls were selected form the Alice Springs and Barkly regions of the NT, and the Kimberley, Pilbara and Goldfields regions of WA. Non-rotavirus notifiable gastrointestinal infections included campylobacter, shigella, salmonella and cryptosporidium, and controls were excluded if they were also identified as a rotavirus case. Age, Aboriginal status, sex and location of residence were obtained from the disease register for inclusion in the regression analysis.

### 2.5. Immunisation Status

The immunisation status of all rotavirus cases, population controls and disease register controls were determined from the Australian Immunisation Register (AIR). Full vaccination was defined as AIR-documented receipt of at least two doses of Rotarix for children living in the NT and at least three doses of RotaTeq for children living in WA. Partial vaccination was defined as AIR-documented receipt of one dose only of Rotarix for children living in the NT and either one or two doses only of RotaTeq for children living in WA. Unvaccinated children were defined as those registered on the AIR, but without documented receipt of any rotavirus vaccines. In circumstances where a child had a vaccine dose recorded as dose two or dose three on the register, but where an earlier dose was not recorded, it was assumed the missing dose had been given [10]. A vaccine dose was considered administered on the date recorded as administered on the register (i.e., without any post-vaccination censoring). A vaccine dose was considered invalid if (1) administered too early (before six weeks of age or <28 days from prior vaccine dose), (2) it exceeded the recommended number of vaccine doses in the schedule (>2 doses of Rotarix or >3 doses of RotaTeq) or (3) the administered vaccine was different to the prior vaccine (mixed Rotarix/RotaTeq vaccination schedule). Children were excluded from selection as cases and controls if they had an invalid vaccine dose. Children were also excluded from the analysis if they were recorded as having received the non-programmatic vaccine for their resident jurisdiction (i.e., RotaTeq but living in the NT, or Rotarix but living in WA).

### 2.6. Statistical Analysis

Conditional logistic regression was used to determine the odds ratio (OR) of vaccination for rotavirus cases compared with matched population controls from the immunisation register. Additional models were fit to compute the OR for any dose of vaccine (full and/or partial vaccination) vs none, full vaccination vs none, partial vaccination vs none, and full vs partial vaccination. Subgroup analyses were by jurisdiction (NT versus WA), and by age (<12 months versus ≥12 months).

For the disease register analysis, ordinary logistic regression was used to determine the odds ratio of vaccination for rotavirus cases compared with disease register controls. Age (months), sex, Aboriginal status (Aboriginal vs non-Aboriginal) and jurisdiction of residence (NT vs WA) were included in the model, together with an interaction term for Aboriginal status and jurisdiction of residence.

Assuming a baseline population vaccine coverage of 80%, we estimated that 80 matched sets of cases and population controls, with 10 controls for each case, would have at least 80% power to detect a significant real-world vaccine effectiveness of 45% (OR = 0.55).

All analysis was performed using Stata, version 15.1 (Stata).

### 2.7. Ethics Committee Approvals

Approval was granted by the Central Australian Human Research Ethics Committee (CAHREC 18-3219), the Human Research Ethics Committee of the Northern Territory Department of Health and Menzies School of Health Research (HREC 18-3248), the Department of Health Western Australian Human Research Ethics Committee (DOH HREC 2018/30), the Western Australian Aboriginal Health Ethics Committee (HREC 891) and the Charles Darwin University Human Research Ethics Committee (H19040). Approval to access data held by the Australian Immunisation Register was granted by the Australian Government Department of Health.

## 3. Results

The rotavirus epidemic occurred between 1 March and 30 June 2017. A total of 194 vaccine-eligible children aged ≥6 weeks were identified as rotavirus cases from which 171 were eligible for inclusion in the study (see Figure 1).

The median age of rotavirus infection was 19 months (range from 1 to 94 months). Most rotavirus cases were among children who identified as Aboriginal and/or Torres Strait Islander (NT 86%, WA 75%). Genotype results were available for only 60% of rotavirus cases, however, of those typed, all were G2P[4] strains. A total of 99 children were documented as having been hospitalised with rotavirus infection—78% of rotavirus cases in the NT and 39% of rotavirus cases in WA. Hospitalisation status was unknown for 15% of WA rotavirus cases (see Table 1).

Among rotavirus cases, 65% were fully vaccinated, 25% partially vaccinated and 10% unvaccinated; among matched population controls from the immunisation register, 71% were fully vaccinated, 21% partially vaccinated and 8% unvaccinated. In the population-based analysis, the odds ratio of receipt of any doses of rotavirus vaccine versus none was 0.79 (95% CI, 0.46–1.34). For the NT and WA, the OR of any doses versus none was 1.10 (95% CI, 0.50–2.41) and 0.56 (95% CI, 0.27–1.16), respectively. For children aged <12 months and for children aged ≥12 months, the ORs were 0.48 (95% CI, 0.22–1.02) and 1.22 (95% CI, 0.55–2.73), respectively. The OR of full versus partial vaccination was 0.65 (95% CI, 0.42–1.01) (see Table 2 and Figure 2).

Of the 171 notified rotavirus cases above, 149 were age eligible for inclusion in the disease register analysis (aged ≥24 weeks or ≥32 weeks in the NT and WA, respectively). A total of 347 vaccine-eligible children were identified as having non-rotavirus gastrointestinal infections in the twelve-month period from 1 January and 31 December 2017. Of these children, 299 were eligible for inclusion (Appendix A
Appendix A). The median age of disease register controls was older than that of rotavirus cases, 29 months vs. 20 months (Appendix A
Appendix A). Disease register controls were less likely to be hospitalised than rotavirus cases (35% vs 58%) and, in WA, were less likely to identify as Aboriginal (41% vs 74%).

In the disease register analysis, 73%, 19% and 8% of cases were fully vaccinated, partially vaccinated and unvaccinated, respectively, compared with 83%, 12% and 5% of controls. The adjusted OR of any doses of rotavirus vaccine versus none was 0.58 (95% CI, 0.24–1.39); for WA and NT children, the adjusted ORs were 0.30 (95% CI, 0.09–0.98) and 1.40 (95% CI, 0.34–5.80), respectively, and for children aged <12 months and ≥12 months old, the adjusted ORs were 0.28 (95% CI, 0.03–2.83) and 0.81 (95% CI, 0.29–2.28), respectively. The adjusted OR of full vs. partial vaccination was 0.63 (95% CI, 0.35–1.13) (see Table 2 and Appendix A
Appendix A).

Additional analyses were performed as requested after peer review—including restricting the population-based analysis to children age-eligible for full vaccination only (aged ≥24 weeks in the NT and ≥32 weeks in WA), restricting the population-based analysis to children aged <5 years and restricting the population-based analysis to Aboriginal children only. An additional analysis was also run without the ‘missing dose assumption’, i.e., in circumstances where a child had a vaccine dose recorded as dose two or three on the register but where an earlier dose was not recorded, the cases and controls were reclassified as ‘partially vaccinated’ (Appendix A
Appendix A). This resulted in the reclassification of 26 population controls as partially vaccinated, but no change to the classification of rotavirus cases. The results of the additional analyses were broadly in keeping with the per-protocol analysis.

## 4. Discussion

In the context of a G2P[4] rotavirus epidemic with 171 laboratory confirmed rotavirus notifications, we failed to find evidence that either rotavirus vaccine provided strong protection against rotavirus gastroenteritis. This contrasts with the large decrease in rotavirus morbidity and mortality observed globally in young children following the licensing of the oral two rotavirus vaccines, Rotarix and RotaTeq, in 2006 [2,7,11].

The 2017 G2P[4] rotavirus epidemic in the Northern Territory and adjoining regions of rural and remote Western Australia predominantly affected Aboriginal and Torres Strait Islander children (NT 86%, WA 75%). Two thirds of cases (65%) were fully vaccinated, and cases were only slightly less likely to have received a vaccine dose than matched population controls sampled from the immunisation register (OR of 0.79 is equivalent to a VE of 21% where VE = 1—OR). There was some evidence of protection among the subgroup of children <12 months old, although all 95% confidence intervals included one (no effect) and there was significant overlap in the confidence intervals across the subgroup analyses. We found little evidence of a protective effect for full vaccination overall (OR of full vs. no vaccination 0.83 (95% CI, 0.43, 1.58)), although there was some evidence that fully vaccinated children were better protected than unvaccinated children in Western Australia (OR of full vs no vaccination for WA 0.40 (95% CI, 0.18–0.93)). We also found some evidence that fully vaccinated children were moderately better protected than partially vaccinated children (OR of full vs. partial vaccination 0.65 (95% CI, 0.42–1.01)). These findings are consistent with recently published vaccine effectiveness studies evaluating the performance of Rotarix in New South Wales and both Rotarix/RotaTeq in Western Australia. In both studies, VE estimates were highest for fully vaccinated children aged <12 months, and there was evidence of increasing vaccine effectiveness with increasing doses of both Rotarix and RotaTeq vaccines [12,13].

Rotarix is a live, monovalent, attenuated oral rotavirus vaccine derived from the most common human rotavirus strain G1P[8], and RotaTeq is a pentavalent (G1, G2, G3, G4, P[8]) human–bovine reassortant vaccine [14]. While post-licensure studies have reported similar vaccine effectiveness levels for Rotarix and RotaTeq [2], very few studies have directly compared the effectiveness of each vaccine in the same setting or during the same outbreak [15,16,17]. While there is good evidence that RotaTeq is protective against G2P[4] strains [18], post-licensure studies have shown mixed results for the effectiveness of Rotarix against G2 strains [8,19] and in some jurisdictions using Rotarix, G2P[4] has emerged as the dominant circulating genotype [20,21,22,23]. An earlier study of a 2009 G2P[4] outbreak amongst NT Aboriginal infants failed to show that the rotavirus vaccine provided strong protection (OR 0.81 (95% CI, 0.32–2.05)) [8]. In our study, all rotavirus samples sent for genotypic analysis from the five administrative health regions between March and June 2017 were identified as G2P[4]. Given the epidemic was well-defined in time and geography, it is reasonable to assume that G2P[4] accounted for all epidemic cases; this study provides a unique opportunity to evaluate the performance of both Rotarix and RotaTeq during the same G2P[4] epidemic and in similar, albeit geographically distinct, populations. While the point estimate of the OR was consistently lower in the jurisdiction using RotaTeq (consistent with better effectiveness), the confidence intervals were wide and overlapping. Small rotavirus case numbers in both jurisdictions and programmatic differences in how cases are ascertained limit our ability to draw conclusions about the comparative effectiveness of the vaccines in this study.

While there was evidence of a protective effect among younger children, our estimates suggest that a strong protective effect of vaccination is unlikely among older children. The median age of rotavirus infection was 19 months with a substantial proportion of cases occurring among children aged 12–23 months (NT 44%, WA 39%). Decreased vaccine protection in the second year of life and persistent burden of rotavirus disease have been reported in other high-burden low-resource settings [2,5,24]. Possible determinants of poor vaccine response include high levels of maternally-derived, vaccine-neutralising anti-rotavirus antibodies, poor infant nutrition, intestinal microbiota imbalance, environmental enteropathy, comorbid infections such as HIV and a high diversity of circulating rotavirus strains [25]. In the population included in this study, children are very unlikely to have been HIV infected, but other infective comorbidities are common. Apart from reduced vaccine-induced protection, programmatic restrictions, including upper age-limits for rotavirus vaccine administration may also diminish the program. An early rotavirus vaccine, RRV-TV, caused intussusception in a small number of vaccinated older infants [26] and despite reassuring phase 3 clinical trial safety results, the manufacturers of Rotarix and RotaTeq have conservatively recommended upper age limits on the administration of their vaccines—24 weeks for Rotarix and 32 weeks for RotaTeq. In practice, this limits opportunity to complete the full vaccination schedule and eliminates the possibility of catch-up of missed vaccinations in later childhood [25]. Delayed and/or incomplete vaccination is more common among Australian Aboriginal children [27] and in one observational study, two-dose DTPa coverage increased by a further 16% after the upper age limit of rotavirus vaccine administration (from 75% to 91% in Aboriginal infants), whereas two-dose rotavirus vaccine coverage increased by only 3% (from 75% to 78% in Aboriginal infants) [28]. This suggests that relaxing the upper age restrictions for rotavirus vaccines, as recommended by WHO for countries with high rotavirus burden [3], could be considered as a strategy for improving vaccine uptake and schedule completion. 

The validity of case-control methods is largely dependent on adequate control of confounders, that is, factors which are causally related to both vaccination and baseline risk of disease [29]. In our setting, vaccination coverage is influenced by age, Aboriginal status, geographical location and calendar time; age and Aboriginal status remain the two strongest baseline risk factors for rotavirus gastroenteritis requiring hospitalisation [7], and epidemics are clustered in geographic space and time. Our study therefore sought to control for these potential confounders by directly matching cases to population controls on age (date of birth), Aboriginal status and location of residence, and by confining the analysis to the defined outbreak period. In the disease register analysis, these factors were not matched but were captured and adjusted for in the regression analysis. This study could not directly measure socio-economic status for individual cases and controls, although Indigenous status and remoteness of residence may be considered surrogate measures, with the Alice Springs, Barkly, Kimberley and Goldfields regions encompassing some of the most socially disadvantaged regions in Australia, as measured by the Index of Relative Socioeconomic Advantage and Disadvantage.

While the jurisdiction-based notifiable infectious disease databases are believed to capture all laboratory-confirmed rotavirus cases during the epidemic, we acknowledge that not all children with rotavirus gastroenteritis present for medical care, are referred for testing, or complete testing when it is recommended. Rotavirus vaccines have been found to be more effective in preventing severe disease requiring hospitalisation than asymptomatic and other less severe forms of infection [2]. While we were not able to directly ascertain disease severity, most cases in this study are likely to have had either moderate or severe gastroenteritis because all sought medical care (in order to be hospitalised), and 78% and 39% were hospitalised in the NT and WA respectively.

It is also acknowledged that the propensity to seek medical care for rotavirus gastroenteritis symptoms may be associated with the propensity to access medical care for other reasons, including vaccination, and this is a potential source of bias in the population-based analysis which may have caused us to underestimate vaccine protection. The disease register analysis is less likely to be affected by this bias because the vaccination status of rotavirus cases was compared to that of other children with (non-vaccine preventable), notifiable gastrointestinal clinical infections, i.e., children with clinical presentations which are likely to have been indistinguishable from rotavirus infection and who also underwent microbiological testing. The results of the disease register nested analysis were limited by small numbers, especially in the subgroup analyses, but were in broad agreement with the population-based analysis.

Rotavirus gastroenteritis cannot be reliably distinguished from other causes of non-bloody diarrhea on clinical grounds, and so only laboratory confirmed cases reported to the notifiable infectious disease databases were included. The sensitivity and specificity for detecting rotavirus in stool samples using commercially available EIA is high, although false positives and false negatives have been reported [30]. This is noted as a limitation of the nested disease register case-control study, where an assay error may result in misclassification of a case as a control, or vice versa, which would have caused us to underestimate vaccine protection.

While the Australian Immunisation Register provides credible individual and population-level data regarding vaccine coverage by vaccine type, date-of-birth, location of residence and Aboriginal status, controls were matched to cases based on their location of residence, as recorded on the register in October 2019, which may or may not accurately reflect their jurisdiction of residence between March and June 2017. It is unclear what, if any, bias this may have caused.

## 5. Conclusions

The incorporation of two rotavirus vaccines into the Australian NIP in 2007 has resulted in a substantial and sustained decrease in rotavirus morbidity across most of Australia, although Aboriginal and Torres Strait Islander children remain at increased risk of severe rotavirus disease requiring hospitalisation [7]. Our evaluation of the 2017 G2P[4] rotavirus epidemic in remote Australia suggests that rotavirus vaccination provided little protection against notifiable rotavirus disease for children living in rural and remote Australia, with the likely exception of children aged <12 months for whom moderate evidence of protection was found.

The admission of an additional 99 children with gastroenteritis to small regional and remote hospitals over fourteen weeks highlights the ongoing public health importance of rotavirus and the need for strategies to better protect Aboriginal children. Our data indicate a likely benefit from full rather than partial vaccination, underscoring the importance of completing the rotavirus schedule. Schedule completion could be enhanced by relaxing the upper age limit of rotavirus vaccination as has been recommended by the World Health Organisation for high-burden settings [3].

Our study also reports a high percentage of rotavirus cases in children aged 12–23 months and decreased vaccine protection among children older than 12 months. It is plausible that administering an additional or booster dose of rotavirus vaccine to slightly older children (beyond manufacturer upper age limit restrictions) may extend protection into the second year of life. Scheduling a third dose of Rotarix vaccine (at between 6 and 11 months old) is currently under investigation in the NT [31].

## Figures and Tables

**Figure 1 pathogens-09-00790-f001:**
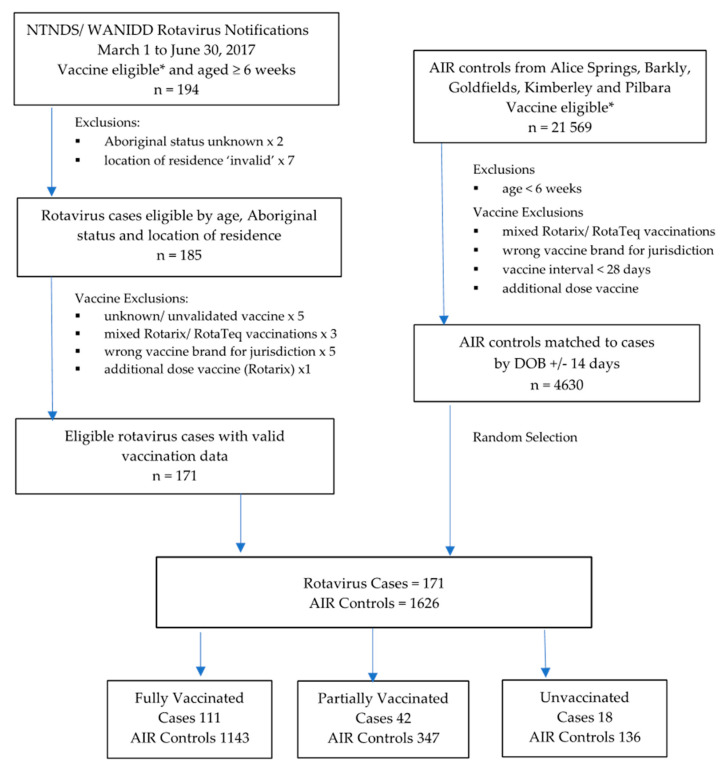
Selection of rotavirus cases and matched population controls from the Australian Immunisation Register. * Vaccine-Eligible: children eligible by date of birth to have received at least one dose of Rotarix vaccine (those born after 1 July 2006 in the Northern Territory) or at least one dose of RotaTeq vaccine (those born after 1 May 2009 in Western Australia).

**Figure 2 pathogens-09-00790-f002:**
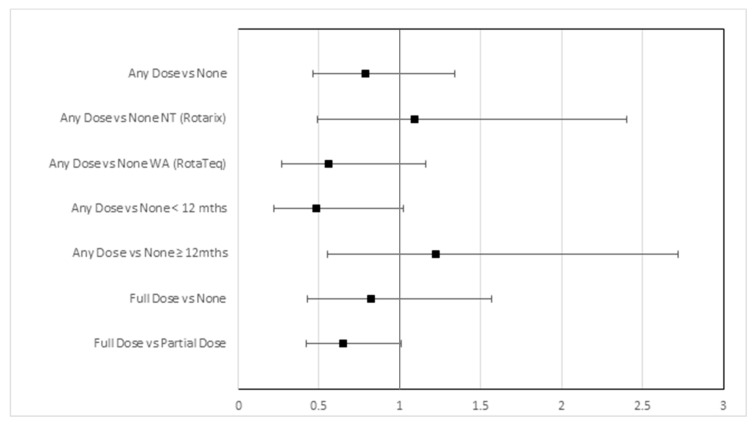
Odds ratio of vaccination in rotavirus cases versus controls in the population-based analysis.

**Table 1 pathogens-09-00790-t001:** Baseline characteristics of rotavirus cases.

Characteristic	Rotavirus Cases	
	NT	WA
	n = 83	n = 88
**Age**		
Median age (months)	18	19
Age range (months)	1 to 72	1 to 94
6 weeks to <24 wks (NT only)	12 (14%)	
6 weeks to <32 wks (WA only)		10 (11%)
6 weeks to <1 year	25 (30%)	23 (26%)
1 year to <2 years	36 (44%)	34 (39%)
2 years to <3 years	11 (13%)	15 (17%)
3 years to <4 years	7 (8%)	5 (6%)
4 years to <5 years	3 (4%)	5 (6%)
≥5 years	1 (1%)	6 (6%)
**Sex**		
Female	42 (51%)	43 (49%)
Male	41 (49%)	45 (51%)
**Aboriginal Status**		
Aboriginal	71 (86%)	66 (75%)
Non-Aboriginal	12 (14%)	22 (25%)
**Location of Residence**		
Alice Springs	70 (84%)	
Barkly	13 (16%)	
Goldfields		17 (19%)
Kimberley		49 (56%)
Pilbara		22 (25%)
**Genotype**		
G2P[4]	44 (53%)	59 (67%)
Unknown	39 (47%)	29 (33%)
**Hospitalisation**		
Yes	65 (78%)	34 (39%)
No	18 (22%)	41 (46%)
Unknown		13 (15%)
**Vaccination**		
0 doses	8 (10%)	10 (11%)
1 doses	15 (18%)	8 (9%)
2 doses	60 (72%)	19 (22%)
3 doses		51 (58%)

**Table 2 pathogens-09-00790-t002:** Odds ratio of vaccination in rotavirus cases versus controls in the population-based analysis and the disease register analysis.

	Immunisation Register Analysis	Disease Register Analysis
Immunisation Status	Cases	Controls	Odds Ratio (95% CI)	Cases	Controls	Odds Ratio (95% CI)
**Any Dose vs. None**	n = 171	n = 1626	0.79 (0.46, 1.34)	n = 149	n = 299	0.58 (0.24, 1.39)
≥One Dose Vaccine	153	1490		137	283	
Unvaccinated	18	136		12	16	
**Any Dose vs. None NT (Rotarix)**	n = 83	n = 753	1.10 (0.50, 2.41)	n = 71	n = 123	1.40 (0.34, 5.80)
≥One Dose Vaccine	75	676		68	114	
Unvaccinated	8	77		3	9	
**Any Dose vs. None WA (RotaTeq)**	n = 88	n = 873	0.56 (0.27, 1.16)	n = 78	n = 176	0.30 (0.09, 0.98)
≥One Dose Vaccine	78	814		69	169	
Unvaccinated	10	59		9	7	
**Any Dose vs. None < 12 mths**	n = 48	n = 449	0.48 (0.22, 1.02)	n = 26	n = 37	0.28 (0.03, 2.83)
≥One Dose Vaccine	37	392		21	36	
Unvaccinated	11	57		5	1	
**Any Dose vs. None ≥** **12mths**	n = 123	n = 1177	1.22 (0.55, 2.73)	n = 123	n = 262	0.81 (0.29, 2.28)
≥One Dose Vaccine	116	1098		116	247	
Unvaccinated	7	79		7	15	
**Full Dose vs. None**	n = 129	n = 1008	0.83 (0.43, 1.58)	n = 121	n = 264	0.55 (0.23, 1.32)
Fully Vaccinated	111	913		109	248	
Unvaccinated	18	95		12	16	
**Full Dose vs. None NT (Rotarix)**	n = 68	n = 529	2.06 (0.62, 6.83)	n = 61	n = 117	1.27 (0.31, 5.23)
Fully Vaccinated	60	469		58	108	
Unvaccinated	8	60		3	9	
**Full Dose vs. None WA (RotaTeq)**	n = 61	n = 479	0.40 (0.18, 0.93)	n = 60	n = 147	0.29 (0.09, 0.96)
Fully Vaccinated	51	444		51	140	
Unvaccinated	10	35		9	7	
**Full Dose vs. Partial Dose**	n = 153	n = 1350	0.65 (0.42, 1.01)	n = 137	n = 283	0.63 (0.35, 1.13)
Fully Vaccinated	111	1060		109	248	
Partially Vaccinated	42	290		28	35

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
