# Peer review of "Retrospective Case-Control Study of 2017 G2P[4] Rotavirus Epidemic in Rural and Remote Australia"

_pathogens, 2020, doi:10.3390/pathogens9100790_

Round 1

Reviewer 1 Report

Thank you for the opportunity to review this interesting paper, which adds to the literature on the potential effectiveness (or lowered effectiveness) of rotavirus vaccines (particularly Rotarix) against G2P[4] and in underresourced populations. I have some comments for the authors to consider.

Introduction

  • Consider adding a bit more information on WA and NT—what are they like? How remote? General observations on healthcare access?

Methods

  • In line 87, please make it explicit that age-eligibility was based on birthdate relative to vaccine introduction
  • In lines 137-141, please clarify what was done with children with “invalid” doses. Excluded from analysis?
  • What units were used for age in the disease register analysis?
    • It might be important to account for age in months or even weeks given the seasonality of rotavirus and other causes of gastroenteritis
  • Did you consider stratified analyses for Aboriginal status? This might be useful to investigate as another proxy for socioeconomic status or healthcare-seeking behaviors.
  • You may consider limiting to children <5 given that older children made up a small proportion of your population and may be somewhat different in terms of vaccination status, AGE risk, and other factors
  • Did you consider limiting the registry-based analysis to the same time period as the epidemic? Or were there too few non-rotavirus cases?

Results

  • You could consider using one of the supplemental tables in your main manuscript so as to show the characteristics of cases vs. controls
  • You might consider also reporting the jurisdiction-stratified full vs. none results in Table 2—to me, it is interesting that in WA, Full vs. none seems to be moderately effective, whereas in NT, there does not appear to be any effect of vaccination.
  • I do not understand why supplemental Table 4 presents both stratified results and results treating the stratifying variable as an effect modifier—to me, these are just doing the same thing (in very slightly different ways). I think you should pick either the stratified or the interaction-term model to present.
  • As mentioned above, I would also consider a stratified analysis by Aboriginal status
  • The age distribution of cases and controls in Supplemental Table 3 is somewhat interesting—I would have expected children from NT to skew older given the broader age eligibility, but the reverse seems to be true. Do you think this is related to differences in Aboriginal status? To other differences in the circulating pathogens?
  • Suggest adding % breakdowns for Geographical Areas and Organisms in Supplemental Table 3

Discussion

  • The second paragraph notes that no significant effects of vaccination were found, but Supplemental Table 4 shows a significant effect of full vs. no vaccination in WA—I would suggest rewriting this paragraph to discuss this finding. I understand that you may not wish to place too much emphasis on it given that Ns were small and overall results did not show significance, but I think it is plausible that you might see an effect of full RotaTeq vaccination but still not find a significant effect of full+partial combined or an effect for Rotarix.

Conclusion

  • Lines 316 – 317 mention “the admission of an additional 100 children”, but this is not mentioned anywhere in the results or earlier in the paper; suggest deleting these specifics or clarifying where this number came from
  • I’m not sure that this particular study suggests the possibility of a third dose or relaxation of age restrictions—maybe if you added some more information about the age at which children were vaccinated, then this could support relaxation of age restrictions. For the second point, about a booster dose, I would suggest emphasizing the possible waning of vaccine effectiveness. I’m also not sure that your study findings suggest Rotarix for a booster dose and not RotaTeq, given that age- and area-stratified results were not presented

Reviewer 2 Report

This article presents a vaccine effectiveness analysis in Australia during a G2P4 rotavirus outbreak in 2017 in 1 area using Rotarix and another using RotaTeq. It uses 2 methodologies to enroll controls. The findings are overstates given about half of the specimens do not appear to have been genotyped and the sample size may be too small.

Introduction: Reference #2 (Jonesteller et. al) was recently updated. The updated version can be sited as:

Burnett E, Parashar UD, Tate JE. Real-world effectiveness of rotavirus vaccines, 2006–19: a literature review and meta-analysis. The Lancet Global Health. 2020 Sep 1;8(9):e1195-202.   Methods:   Lines 134-136: I'm wondering about the decision to include doses not recorded in the immunization registry (which you state covers 99% of healthcare encounters) while excluding mixed series and children vaccinated with the vaccine not given in their state. It seems like this could misclassify children as fully vaccinated when they are in fact partially vaccinated. At a minimum, you should consider a sensitivity analysis excluding these children.   Age limits: It's possible that some of the differences in OR point estimates between these two populations may be due to using different inclusion criteria for the lower age limits. It would also be helpful to show in Table 1 the number of children 6weeks-24 weeks. Consider limiting both analyses to the age when children would have been eligible to have received the completed series. You also included a few children up to 7 years old. The overall results would be more comparable to other studies if limited to <5 years   Sample size: please state you assumptions and estimated sample size. I suspect the number of enrolled  cases may be too small and this may partially explain the wide confidence limits.   Lines 136-138: It is not clear how "invalid" doses were considered. They appear to be included in Table 1. You also are mixing 2 types of invalidity which could have implications for your analysis: shortened interval, which may reduce the effectiveness, and extra doses which may improve the effectiveness. Additionally, it is not clear if there was a minimum amount of time between vaccination and illness onset before the dose was considered "administered".   Results: In table 1, it shows that about half of specimens were G2P[4] and the other half were unknown. Were they not tested at all and therefore could also be G2P[4]?   There appears to be a huge difference in performance between the two states-- this is not addressed anywhere in your manuscript. The results in WA are significant in some analyses while NT show increased risk with rotavirus vaccination.   You might considered adding full v. non by age group and partial v. none instead of full v. partial.   Discussion:   You skate around that you have no measure of disease severity. Rotavirus vaccines have been found to be most effective in severe disease. This should be included in the limitations.  

Round 2

Reviewer 1 Report

All comments have been addressed. My one remaining suggestion is to make some reference to the timeliness of non-rotavirus vaccinations in these regions--this may support your point about relaxing the rotavirus age restrictions. For instance, if children tend to complete hexa/pneumo late, perhaps finishing after 8 months of age, then this would argue for extending rota vaccine eligibility.

Author Response

REVIEWER ONE: All comments have been addressed. My one remaining suggestion is to make some reference to the timeliness of non-rotavirus vaccinations in these regions--this may support your point about relaxing the rotavirus age restrictions. For instance, if children tend to complete hexa/pneumo late, perhaps finishing after 8 months of age, then this would argue for extending rota vaccine eligibility.

RESPONSE TO REVIWER ONE; thank you, we have now added the following to the Discussion see Line 292 'Delayed and/or incomplete vaccination is more common among Australian Aboriginal children (27) and in one observational study, two-dose DTPa coverage increased by a further 16% after the upper age limit of rotavirus vaccine administration (from 75% to 91% for Aboriginal infants), whereas two-dose rotavirus vaccine coverage increased by only 3% (from 75% to 78% for Aboriginal infants) (28). This suggests that relaxing the upper age restrictions for rotavirus vaccines, as recommended by WHO for countries with high rotavirus burden (3), could be considered as a strategy for improving vaccine uptake and schedule completion. 

Reviewer 2 Report

Comments have been adequately addressed.

Author Response

REVIEWER TWO COMMENTS: Comments have been adequately addressed.

RESPONSE TO REVIEWER TWO: thank you, no further changes required.